# Challenges to Systems of Long-Term Care: Mapping of the Central Concepts from an Umbrella Review

**DOI:** 10.3390/ijerph20031698

**Published:** 2023-01-17

**Authors:** Luís Filipe Barreira, Abel Paiva, Beatriz Araújo, Maria Joana Campos

**Affiliations:** 1Center for Interdisciplinary Research in Health, Instituto Ciências da Saúde, Universidade Católica Portuguesa, Rua de Diogo Botelho 1327, 4169-005 Porto, Portugal; 2Instituto de Ciências da Saúde do Porto, Universidade Católica Portuguesa, R. de Diogo Botelho 1327, 4169-005 Porto, Portugal; 3Porto School of Nursing, Escola Superior de Enfermagem do Porto, 4200-072 Porto, Portugal

**Keywords:** long-term care, integrated care, home care

## Abstract

The ageing of the population poses urgent challenges to the health and social protection sectors, including the need for greater adequacy and integration of health care services provided to older people. It is considered necessary and urgent to understand the state-of-the-art of community-based models of care for older people in institutional care and at home. This study aims to map the concepts that politicians and providers need to address through an umbrella review as a review method. Articles describing the structuring aspects of care models appropriate to the needs in long-term care and systematic reviews or meta-analyses targeting people aged 65 years or more were considered. A total of 350 studies met the inclusion criteria and were included in the review. The results identified the need to contribute to effective and more efficient integration and articulation of all the stakeholders, based essentially on professional care at the patient’s homes, focused on their needs using the available technologies, empowering patients and families. Eight categories emerged that addressed factors and variables involved in care models for the long-term care needs of institutionalised and home-based older people as a guarantee of accessibility to healthcare and to enhance the well-being and quality of life of patients and family caregivers.

## 1. Introduction

In recent decades, ageing has occurred faster than projected, with older adults representing a growing proportion of the population. In many developed countries, an inversion of the population pyramid was verified [1].

Population ageing implies a greater need and demand for healthcare, with health expenditure becoming particularly expensive in the last year of life. The verified changes determine the need to rethink society’s responses to this population. The health systems must be rethought to enhance and increase their capacity to respond to the different and growing health challenges evidenced by this age group, providing appropriate care to people’s needs and wishes, with the required quality and safety in a more cost-effective manner [2].

Many countries have hospital-based health systems. Recent results show that this model of care does not allow a comprehensive and adequate response to the arising needs from the demographic and epidemiological transition that has characterized recent decades [3] and requires an effective, articulated response between the different levels of care.

This incapacity is the result of a set of diverse factors, among which it has highlighted the greater diversity of care required by the older adult population. The increased fragmentation of care due to differentiation and, finally, the increase in private providers and the consequent impoverishment of economically and financially weaker population groups reveals this inefficiency [4].

It is now recognized that the function and responsibility of health systems are not confined to providing care but also play an important social role, integrating social and financial protection solutions against the costs of illness and associated inequalities. Particularly concerning individuals or population groups in a situation of greater dependency or vulnerability [1,2].

As referred in the Public Finance Council’s 2022 report, it is estimated that around 7% of the world’s population has long-term care needs and that this figure is 19% in the population aged 65 and over, according to the European Commission’s 2021 Ageing Report, Economic & Budgetary Projections for the EU Member States (2021) [5] (p.17).

The fragility of the social protection systems for the older adult and the evidence that they are fragmented and inadequately monitored were evident during the COVID-19 pandemic. The absence of an integrated and coordinated perspective contributed to the uncontrolled spread of the infection and devastating mortality rates among the older adult population, mostly living in nursing homes, in addition to the significant impact on the well-being and health of users, carers, families and their communities [6].

The disproportionate impact of the recent pandemic among older people, including institutionalised older people, in terms of hospitalizations and deaths, has emphasized some of the urgent challenges that population ageing poses to the health and social protection sectors [6], including the need for greater appropriateness and integration of the health care provided.

As we wrote before, the global population is ageing faster and faster. Long-term care interventions are needed to deal with the increasing prevalence of chronic diseases, with the loss of functional capacity and self-care abilities.

The increase in dependency and chronic diseases will result in a decline in the global population’s capacity for self-care and functionality, creating significant challenges in all sectors of society, mainly in health and social care. These challenges become more outstanding and demand a response from governments due to the lack of sufficient and/or adequate services and the costs associated with adequate care. This trend will increase demand for adequate services and require immediate responses from health systems.

The COVID-19 pandemic has become further evidence of the need to reform and invest in long-term care systems.

Long-term care is a range of services in various settings such as nursing homes, assisted-care homes, long-term care facilities, home, residential homes for the elderly and in the community [7].

The European Commission defines long-term care as “a range of services and assistance for people who, as a result of mental and/or physical frailty and/or disability over an extended period of time, depend on help with daily living activities and/or need some permanent nursing care. These daily living activities may be self-care activities such as bathing, dressing, eating, getting in and out of bed or a chair, moving around, using the toilet, and controlling physiological functions. Alternatively, they may be related to independent living activities such as preparing meals, managing money, shopping for groceries or personal items, performing light or heavy housework and using a telephone” [8].

The European political agenda and numerous international agencies recognize that there are models of care focused on the needs of care of this population group. As there are already known data regarding the pandemic period, it was considered necessary and urgent to understand the state-of-the-art services regarding community-based care models for older people.

This conceptual mapping, carried out using the umbrella review methodology, aims to identify and describe structural aspects of integrated care models, presented as appropriate to the growing needs in long-term care focused on older people.

This study intends to map the scientific evidence generated through systematic reviews and meta-analysis, which would facilitate inferring the domains and categories of scientific evidence related to the appropriate care models for the long-term care needs of institutionalised and stay-at-home older people.

## 2. Methods

The umbrella review methodology was used as a review method to map the existing research in the area of long-term care models for older people. Umbrella reviews are reviews of previously published systematic reviews or meta-analyses. They consist of repeating meta-analyses following a uniform approach to all factors to allow its comparison. Among the different types of review, they represent one of the most extensive evidence synthesis methods currently available.

In healthcare, umbrella reviews are essential to map the best available evidence on a given topic. As Aromataris and colleagues (2015) [9] revealed, umbrella reviews provide a starting point for healthcare decision-makers to understand a broad area of knowledge. They also provide health professionals with a comprehensive overview of a topic, helping to lay the foundations for evidence-based practice.

This study presents the concept mapping designed according to the Joanna Briggs Institute’s methodology for umbrella reviews. Considering that the area under study is not a clinical item but a broad theme, the umbrella review was used to synthesize the variables involved in the care models appropriate to the long-term care needs of older people. We used broad terms in the search, such as “home care” and “long-term care”, to not exclude any relevant study that would contribute to the intended mapping.

This umbrella review included articles describing the state-of-the-art models for delivering care in the community for older people. Due to the number of available studies, clinical contexts, and different kinds of patients, the data was organized by categories. The Preferred Reporting Items for Systematic Reviews and Meta-Analyses (PRISMA Statement) checklist [10] used a meta-analysis guideline to report the results of the study. A protocol for this review was published in PROSPERO, 20 September 2020, registration number CRD42020202192.

### 2.1. Inclusion Criteria

The inclusion criteria indicate the basis on which the available reviews were considered for inclusion in the umbrella review and, more importantly, served as a guide for the reviewers to decide on the reviews to be selected during the study selection phase of the review umbrella.

Included studies are exclusively systematic reviews and meta-analyses, not primary or original research. Reviews that incorporated theoretical studies or published opinions as their primary source of evidence were not included in the review, having been considered as one of the exclusion criteria.

Broad selection criteria were used to be comprehensive. Articles were included that: (a) described structuring aspects of care models in long-term care in the community; (b) addressed variables or factors involved in care models suited to long-term care needs; (c) dealt with medical/nursing and other professional services and care aimed at alleviating/reducing pain and suffering, reducing or controlling deterioration in health status in patients with a long-term degree of dependence, implementation of care plans, as well as end-of-life care; (d) contained peer-reviewed systematic reviews and meta-analyses written in English and (e) were targeted at persons age 65 and older.

The defined criteria were developed to guide the selection of articles. Studies were not excluded based on year or publication status. Studies were excluded because: (a) they were not systematic reviews and meta-analyses (e.g., systematic review protocols/workshop/thesis/chapters); (b) did not include relevant data on structuring aspects or variables or factors involved in care models in long-term care in the community; (c) did not meet the age requirement; and (d) were not focused at family caregivers and patients. The defined criteria were developed to guide the selection of articles.

### 2.2. Identifying the Research Questions

The following research question were identified:

What domains and categories of scientific evidence are currently available on appropriate care models for the long-term care needs of institutionalised and at-home older people?

### 2.3. Identifying Relevant Studies

A comprehensive search strategy was developed and refined in collaboration with our research team.

A search was conducted on 24 February 2020 using the following electronic databases: CINAHL^®^ (Academic Search Complete; Business Source Complete; CINAHL Complete; CINAHL Plus with Full Text; ERIC; Library, Information Science & Technology Abstracts; MedicLatina; MEDLINE with Full Text; Psychology and Behavioural Sciences Collection; Regional Business News; SPORTDiscus with Full Text), Scopus e WEB OF SCIENCE. The search was conducted according to the syntax and indexing terms appropriate to each database. The search strategy is included in Table 1.

The open period was selected due to the importance of knowing every piece of evidence available in this field to screen all possibilities.

### 2.4. Study Selection

The included studies focused on healthcare models appropriate to the needs of older people in long-term care, which determined the inclusion of articles addressing long-term care at home, in institutions (nursing homes, assisted living facilities, etc.) and both contexts. The clients were considered both the patient and the family caregiver.

### 2.5. Charting the Data

Identified articles were downloaded into Endnote, where duplicates were removed. Article titles and abstracts were randomly screened independently by two different reviewers and labelled with the following categories: “include”, “exclude”, or “potentially include”. A third reviewer resolved conflicts. Studies included in the full-text review were screened independently by two authors (LFB and MJC), and conflicts were resolved by discussing each article individually.

A data extraction template was created to extract relevant information from the included articles. This template was refined through feedback from our research team. The following data categories were extracted for each article, when applicable:

Author(s), study location, publication details, issue, study design, research aim, sampling method, sample size, data source, data analysis techniques, setting, clinical outcome of interest.

Two reviewers also completed data extraction for each article, where the second reviewer confirmed and completed the results as needed.

### 2.6. Collecting, Summarizing and Reporting the Results

The number of articles included in this review were summarized in Table 2.

## 3. Results

A total of 1251 articles were retrieved from databases, as illustrated in the PRISMA flowchart diagram (Figure 1). Altogether, 357 duplicates were removed, and 183 articles were excluded through the title and abstract screening process. A total of 711 full-text articles were retrieved and evaluated. Ultimately, 350 studies met the inclusion criteria and were included in the review.

We categorized the reported studies into the patient outcomes. The client was divided into the patient, family caregiver, or both. We also categorized the setting into three items: domiciliary (home) context, hospital and nursing home/hospice.

With the 350 articles included in this umbrella review, we drew a conceptual scheme of the reviewed articles to know and map the existing research in this area, as shown in the following figure.

It is essential to explore the core concepts of this mapping. When we analysed the articles, the context of care is often a variable in the reports (contexts mentioned in Table 2 and Figure 2). Without institutionalization refers to the context where people live at home and benefit from long-term care in the community or home. When we refer to home, we talk about the patient or family caregiver’s home. With institutionalization refers to the context where clients stay in long-term care facilities or nursing homes. Research that reports results in the two settings denominate mixed (with and without institutionalization). The care settings in the community, namely with institutionalization, without institutionalization and mixed environments. The client varies in different perspectives. We have articles with various clients in different contexts, as shown in Table 2.

The aspects being considered in older adult care are multifactorial. Thus, the client may be the patient and/or the family caregiver. With regard to the patient, we found articles that focused on bedridden people and others on people, although self-care dependent, who still walk and/or leave their homes.

From the point of view of context and client analysis, eight categories emerged: from hospital to home, case management, home hospitalisation, quality outcomes, interventions, system reform (cases), system reform (financial impact) and technologies.

The definition of the concepts of this mapping is fundamental to the development of this research. The context of care is related to the place where care is developed. In this case, the context is the community. This concept has a connection to Primary Health Care, which the WHO has defined since the Alma-Ata conference as “addresses the main health problems of the community, providing promotive, preventive, curative and rehabilitative services accordingly” [11]. Is essential health care made universally accessible to individuals, families and groups in the community? It forms an integral part of the country’s health care system, which is the nucleus of the community’s overall social and economic development. It is the first contact of individuals, family and the community with the national health care system, bringing health care as close as possible to where people live and work, and constitutes the first element of community health. When referred to the non-institutionalised setting, it is meant the primary health care setting, where home care is included.

Regarding the context of institutionalisation, it is related to care provided in residential structures for the older adult or in long-term care institutions. “Long-term care facilities (LTCFs) are a key sector in providing long-term care for older persons and people with disability. Although the term LTCF has no specific definition, LTCFs in general provide housing, nursing, and supportive services to persons who cannot live independently” [12]. Nursing Homes according to the Instituto de Segurança Social (Welfare Institute) are defined as “a social response, developed in equipment, aimed at collective accommodation, in a context of “assisted living”, for people of an age corresponding to the established retirement age, or others at greater risk of loss of independence and/or autonomy [13].”

Mixed context considers situations in which, in the reviews analysed, primary health care contexts provided at home and care in long-stay units and nursing homes are included.

Before referring to the findings of this review, it is important to mention that we focused the results on the categories that emerged, regardless of context or type of client.

The first category is called from hospital to community, therefore referring to the WHO (2016) concept of transition of care defined as “the various points where a patient moves to, or returns from, a particular physical location or contact a health care professional to receive health care. Includes transitions between the home, hospital, residential care settings and consultations with different health care providers in out-patient facilities” [14].

In this category, we found articles about the functional benefits of high-risk patients after hospital discharge when there was a home assessment carried out by nurses [15]. Another study shows some of the predictors for the transition of care settings, namely that the need for long-term institutional care is associated with age, female gender, dementia, cognitive impairment and functional dependence [16].

There are 71 reasons older people want to remain in their own homes when they may leave the hospital or long-term care/ nursing homes [17]. Some factors are associated with this transition experience, namely the preparation, the communication quality, the quality of care, the family involvement and the roles of the family [18]. Post-discharge interventions, with care provided exclusively at home, reduced hospital stays and improved patient satisfaction. The most effective interventions to prevent inappropriate readmission to hospitals and promote early discharge included integrated hospital-community systems, multidisciplinary service provision, individualized services, hospital-initiated discharge planning and specialized follow-up [19].

Case management can be defined as: “a collaborative process of assessment, planning, facilitation, care coordination, evaluation and advocacy for options and services to meet an individual’s and family’s comprehensive health needs through communication and available resources to promote patient safety, quality of care, and cost-effective outcomes” [20]. Case management programmes (education and counselling services, caregiver training, medication treatment and management, crisis interventions, client empowerment and client advocacy) appear to benefit the frail older adult population. They contribute to a reduction in healthcare utilization and also a statistically significant reduction in the risk of long-term care placement.

Case management impacts patient and carer outcomes. The client outcomes impact mortality, physical or cognitive functioning, clinical conditions, behavioural problems, unmet care needs, psychological health or well-being, and satisfaction with care. For the caregiver, the impact is especially on decreased stress or burden, satisfaction with care, psychological health or well-being, and social consequences, such as social support and client relationships [21]. Integrated care programmes with a nurse case manager impact the improvement in activities of daily living, instrumental activities of daily living, cognitive function and patient satisfaction, as well as the improvement in quality of life and global health perception. Strong positive evidence was found on the avoidable reduction in admissions to acute services and risk of institutionalisation, as well as on better access to social and health services [22]. Case management benefits clients and reduces costs associated with healthcare, helping to improve function and appropriate use of medicines, increase the use of community services, reduce nursing home admissions and increase satisfaction with healthcare [23].

Home hospitalization refers to differentiated care at home provided by a hospital team in cases of acute illness of an older adult who, due to family and social support, can remain at home and does not require hospital care [24]. In the case of acute illness, home hospitalization influences cost reduction in several areas: the unit costs of disease-specific hospital days, the decreasing intensity of care over an admission reflected in the costs of hospital days and the costs of informal care included. However, on the other hand, the evidence that early discharge brings economic benefits related to reduced length of hospital stay and better health outcomes is still limited [25].

The quality assessment in health entails the knowledge of the relationship between structure, processes and results [26]. The outcome, as the name implies, measures the outcome of the effect of healthcare provision, including health-related quality of life and client satisfaction [26]. The indicators are tools that allow measuring the performance (...) aiming to discover certain basic information for decision making in order to improve quality [27].

Regarding the quality outcomes and the caregivers of patients who stay at home, mental health, physical health, involvement in personal and social life, overload, stress, depression, support, preparation for the role and knowledge of self-management are usually assessed. The negative impact on overall health has a higher expression in women, married people and those who care more intensively. Long-term care use is associated with depression, stress and caregiver burden. Uncertainty about the future and the prospects of death, the physical and emotional burden of caregiving, experiencing a limited life, redefining the relationship with the person cared for, and valuing the importance of the supportive environment seems to have a significant impact on the implications on personal life. The strategies with impact include: recognizing the transition, restructuring daily life, maintaining balance in family relationships, assuming the responsibility for care, using social support, and acquiring caregiving skills. Caregivers need practical skills training, fear social isolation, wish to relate to other caregivers for social or learning purposes, and desire respite care [28,29,30,31,32,33].

The quality outcomes related to patients frequently assessed our quality of life, self-management of chronic disease, diabetes, satisfaction with care, relationship with professionals and aspects of improvement in the condition and health/disease [34,35,36,37].

Regarding the interventions category, we can verify that the studies refer to multidisciplinary interventions, specifically, nursing, doctors, nutritionists, psychologists and physiotherapists. However, most approach nursing interventions consider the needs of people in long-term care. Germany, Hungary and Switzerland have a larger supply of nurses in long-term care than other workers. In certain countries, long-term care professionals only sometimes have sufficient knowledge and training, which can seriously affect the quality of care provided [1]. In other OECD countries the proportion of nurses in long term care is greater than other professionals.

Long-term care interventions incorporate support in activities of daily living, the monitoring/follow-up of health status, implementation of care plans, maintenance of records regarding health status, response to treatment, prescription, emotional support, and end-of-life care. These are central factors in the nursing professional and nursing intervention is considered the most routine care patients and caregivers need in long-term care.

Nursing interventions can be defined as any treatment, based on clinical judgment (diagnosis) and knowledge, that the nurse performs to improve patient outcomes [38]. Nursing interventions respond to the identified diagnosis or health problem and are intended to produce the expected outcome. Examples of caregiver-oriented intervention are “respite care”, with positive results in the caregiver’s mental health. Role-matching interventions allow keeping the patient at home but have no impact on the family’s functionality. However, they increase knowledge about the disease, develop problem-solving skills, facilitate social support, and increase caregiver competence. However, they have no significant effect on improving the caregiver’s physical health and little effect on alleviating the caregiver’s burden [39,40].

Patient-centred interventions are of various types. Direct health care impacts the older person’s sense of autonomy, both in decisions and in care and daily life, as well as having an impact on their self-care [41]. Autonomy is strengthened by the older adult’s capacity and the respectful conduct of health professionals, and the involvement between the older adult, caregivers and health professionals is a negotiated interaction with a positive effect on the patient. Interventions focusing on nutritional status and physical activity have positive implications for patients’ health [42,43]. Rehabilitation at home has insufficient evidence of the expected effects on the patient [44]. However, other studies have shown that in the home setting, a more structured action can highlight cost reduction and be promising with regard to the impact on the patient [45]. Health promotion has a positive impact on mortality and some likelihood, when carried out in the home and led by nurses, of offering cost savings and clinical benefits [46]. Standard variables in home-based interventions for non-walking and bed-bound patients included dignity, respect, and choice of care setting [47].

Regarding the system reform category, it included articles that addressed changes in health policies to achieve the “ideal” system. Two subcategories were created, considering the existence of studies that addressed specific cases in some countries and others that addressed the financial impact of system reforms.

In Canada, the home care programmes analysed were mainly less costly. However, they include mixed results and indicate that the cost-effectiveness of home care programmes is an important area for further study [48].

When addressing the alternatives to acute hospitalisation for older people, many options available are safe and appear to reduce resource use. However, the crucial characteristics of older patients for whom the decision to hospitalise is uncertain are age over 75 years, comorbidities/multimorbidities, dementia, home situation, social support and individual skills [49].

A review addressing the effectiveness of integrated care, home hospitalisation, nurse “navigator”, or interventions to improve self-care reduced the readmission rate in older adult patients in Spain [50]. In the UK, analysis of the effect of integrated care or coordination between health services or between health and social care indicated the improved quality of care, increased patient satisfaction, and improved access to care [51].

In line with the above, in Canada, research into the research programme to integrate services to maintain independence has provided evidence that integrating care for older people is beneficial for individuals, reducing the incidence of functional decline and levels of disability and improving feelings of empowerment and satisfaction with care. The research also demonstrated benefits for the health system, with more appropriate use of emergency services and a decrease in consultations with specialist doctors [52].

We can see that overall, integrated, person-centred care is more effective than fragmented care. It meets the needs of the patient and carer, improves functional outcomes, reduces the length of stay, and positively impacts hospital admission rates and patient satisfaction, but not on mortality [53,54,55,56,57,58]. In mental health, the integrated care model is more cost-effective than standard treatment [59]. Factors for successful integrated care include the personal relationship between leaders of different organisations, planning of activities, management and financial resources, support for staff in new roles and organisational and staff stability [60].

Regarding the institutionalisation of the older adult, most of the older adult population in Iran do not have a positive attitude towards nursing homes. They believe that staying and living in nursing homes is a rejection of society [61].

When analysing the effect of organizational and financial health system reforms on the quality of care in high-income countries, the evidence base suggests that privatization and marketization of health systems do not improve quality. Most financial and organizational reforms have inconclusive or adverse effects [53].

On the other hand, in countries where public investment in health policy is traditionally substantial and long-term care has emerged more recently, investment has been used to maintain or increase the informal care available to contain costs [62].

Evidence shows positive cost-containing effects in the design and development of formal long-term care systems, based on three key pillars: (a) a focus on older or more dependent older people, (b) expansion of financing based on individual contributions, and (c) promotion of home care and financial benefits for care in specialized centres (nursing homes and similar establishments) [63].

The evidence is limited regarding the cost-effectiveness of integrated preventive care for frail older people living in the community [57]. However, essential strategies for integrated care for this population were identified. Patients and carers highlighted the continuity of care with a professional they could trust, while providers emphasized better coordination of care between different sectors and services. The perceptions of the integrated care intervention and the target population, the organizational factors of the service and the system-level factors are perceived as facilitating the implementation of integrated care. In contrast, the complexity of care needs, the difficulties with access to the system and the limited involvement of the patient and caregiver in care decisions are presented as factors hindering its implementation [64].

Multidisciplinary care teams, care planning, case management, geriatric assessment, appropriate information technology, changes in organizational culture, and uses of financial incentives are considered fundamental mechanisms for care integration. Evidence suggests that bringing the diverse services that older people with multiple chronic conditions need into a single one would make integrated care easier [65,66]. Person-centred care is fundamental to integrated community-based care, articulated with all levels of the health care delivery system [67].

The interventions of specialist nurses in patients with palliative care needs may effectively reduce the use of specific resources, namely at the level of hospital admissions and re-admissions, length of stay and health care costs. However, there is mixed evidence on the cost-effectiveness of such care [68].

Integrated care programmes for the management of chronic diseases (respiratory, cardiovascular, diabetes and musculoskeletal diseases) have the potential to be cost-effective, achieving more significant health benefits and are cheaper than usual care. The reduction in inpatient and outpatient admissions was the main contributor to cost savings [69]. On the other hand, cost-effectiveness studies on integrated stroke services suggest that they can reduce costs. Specifically, discharge costs with early support are lower than for conventional care, with similar health outcomes [70].

According to the World Health Organization [71], digital health is the field of knowledge and practice associated with developing and using technologies to improve health. Digital health is often used as a broad umbrella that encompasses eHealth but also encompasses other uses of technologies for health, such as the internet, artificial intelligence, big data and robotics. The use of technologies, telehealth or remote patient monitoring has a short-term impact on some indicators. As decreased heart failure-related mortality within 180 days, reduced risk of mortality from all causes of disease, and reduced relative risk of heart failure-related hospitalizations contribute to safety, security and ageing at home [72,73]. Outcomes of telehealth-based services are generally comparable to outcomes of services provided in person; patients preferred a combination of telecare and traditional care, i.e., they should be used as a complement, and privacy is not seen as an issue.

Regarding technologies, results show that medication adherence did not differ. It was not statistically different from standard home care for the quality of life, psychological well-being, physical function, anxiety, depression, disease-specific outcomes or bedside care days at 3, 6, 9 and 12 months after follow-up [74]. Some devices do not work outside the home. A lack of understanding may make it challenging to use telecare correctly.

## 4. Discussion

The umbrella review methodology allowed mapping concepts, organised into categories, contributing to the identification of variables and multiple structuring factors that influence care models suited to the needs of older people in long-term care, institutionalised and at home.

Each of the factors identified relates to the patient and family caregivers, their preferences and needs, the organisation of the health policies in place and the sustainability of health systems.

The complexity of this approach also makes it possible to identify common aspects on which those involved, in particular policymakers, health professionals and society, should focus to make adequate long-term care provision for older people a reality.

This review’s results confirm some aspects already mentioned in other studies [1]. In the sense that older people want to stay at home, as well as confirming that integrated care and case management methodologies carried out by nurses positively impact both the caregiver’s and the patient’s health. Some studies show their economic effectiveness, contributing to an effective cost reduction. Interventions in home settings should be clearly and precisely defined and include the best evidence so that quality indicators positively impact sustained and informed decision-making processes.

A significant contribution to dependent older people is to have access to professional care at home, with dignity, improving their quality of life at this vulnerable stage. The studies highlight the need for this care to be provided by professionals, adapting the current models of care and not through the transfer of caring competencies to family caregivers, who assume the provision of care which, by its nature, should be provided by professionals. Professional care is a determining factor in the issues under study, particularly regarding the expected positive impact, not only on the patient’s health and well-being indicators but also in reducing the overload, stress and implications in the caregivers’ personal and professional lives, with the consequent reduction in negative impacts on the caregivers’ health and well-being. In the same way, the results allow us to define new care models based on examples and success stories in some countries.

The results highlighted the need for the organisation and care management policy to integrate all sectors and stakeholders right from its design. The effective integration and articulation of all those involved more efficiently, based essentially on professional care at the client’s home, using the available technologies, empowering patients and families for their benefit, and contributing in the same way to the sustainability and functioning of the health system and services.

Integrated care also emerges as an essential and central concept, focused on the needs of older people, community-based, recognising the urgency of its implementation at all levels of the health care delivery system, namely through training of the health professionals involved. It is essential to highlight that making the services that older people with multiple chronic conditions need available in a single structure would make integrated care easier.

The study contributes to building appropriate care models worldwide for the long-term care needs of older adults. We can systematize the core ideas that emerged in this paper and add critical elements to the overall research knowledge in this field. The results show the contribution of these care models in guaranteeing access to healthcare, reducing inequalities and asymmetries, and enhancing the well-being and quality of life of patients and family caregivers. These models of care are associated with social, labour and economic impacts, with the possibility of contributing to more sustainable and resilient health systems focused on the wishes and needs of the population.

The coordination and integration of health and social services have a significant benefit in improving the quality of care, access to care, patient and family caregiver satisfaction, the reduction in the functional decline and levels of incapacity of older people, the reduction in the use of emergency services and medical consultations with a specialist.

Person-centred care and integrated care are more efficient and cost-effective in the area of mental health and the management of chronic diseases. They assume one of the most critical aspects of care for the older person, being essential for policy makers in defining policies and health services suited to the needs of older people. Promoting home-based care allows for cost containment in implementing formal long-term care systems.

There are essential variables in the implementation of integrated care, the coordination of care between different sectors and services, the existence of multidisciplinary care teams, care planning, continuity of care with a reference professional (case manager), technologies of adequate information, the organizational culture and the use of financial incentives.

The study contributes to building appropriate care models worldwide for the long-term care needs of older adults, focusing on all areas addressed in this umbrella and mapped in eight categories.

## 5. Conclusions

The conceptual mapping allowed us to identify key concepts and factors in long-term care models for older people. Care models are a broad theme and not a clinical category, so the multiplicity of factors and their implications recommend developing further research on each dimension. Eight categories emerged from the available scientific evidence that addressed factors and variables involved in care models for the long-term care needs of institutionalised and home-based older people. The research implications allow us to understand how we can assess, monitor and adjust long-term care models for older people to the realities of each country and each culture. Implementing this contributes to obtaining health gains and making or influencing public health policies based on the best evidence allowing international comparability, either at the level of resource allocation or regarding the quality and appropriateness of the health care provided.

The study’s limitations result from the inclusion of studies primarily conducted in the United States and Canada, and several articles with low or weak evidence. Another limitation is that we need to progress to the verification of the levels of evidence of the research. Since the objective of the study was to understand which domains of scientific evidence influence the appropriate care models for the needs of older people in long-term care, institutionalised and at home, through conceptual mapping.

The positive aspects of the review include that the research carried out is comprehensive, transparent and follows the methodology adopted, both as to the inclusion criteria and analysis procedures and the synthesis of the results found.

## Figures and Tables

**Figure 1 ijerph-20-01698-f001:**
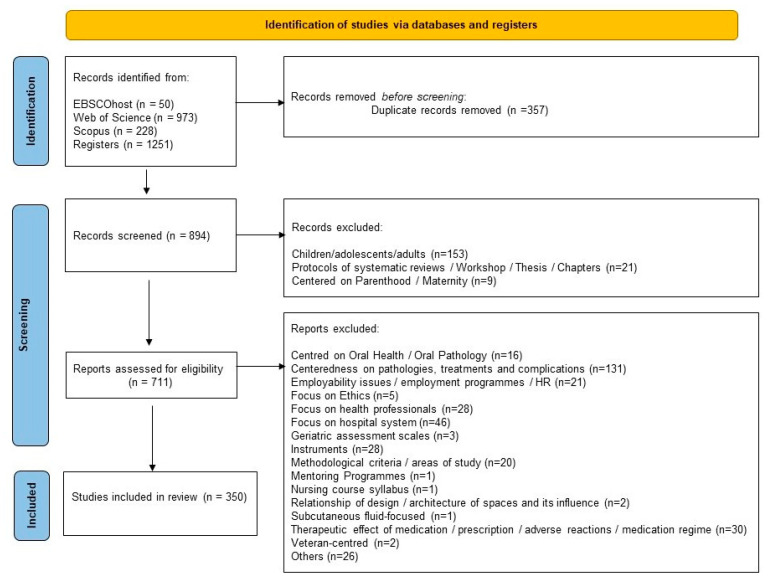
PRISMA 2020 flow diagram.

**Figure 2 ijerph-20-01698-f002:**
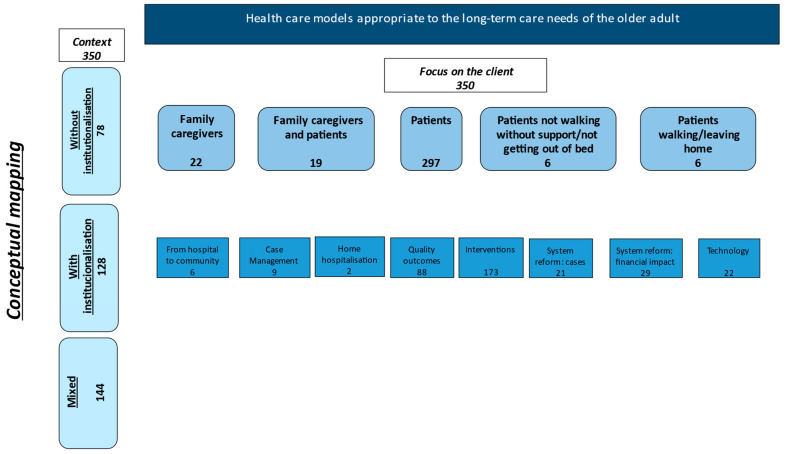
Conceptual mapping scheme.

**Table 1 ijerph-20-01698-t001:** Databases and boolean phrases used.

Interface—Research databases *EBSCOhost* Search Screen—Advanced Search Database—*Academic Search Complete; Business Source Complete; CINAHL Complete; CINAHL Plus with Full Text; ERIC; Library, Information Science & Technology Abstracts; MedicLatina; MEDLINE with Full Text; Psychology and Behavioral Sciences Collection; Regional Business News; SPORTDiscus with Full Text* Limiters—Academic journals (reviewed by experts) Expanders—Apply equivalent subjects Restrict by Subject Age—*aged: 65 + years* Search Modes—*Boolean/Phrase* *(“long-term care” or “integrated care” or “home care”) and “systematic review”)*
*Web of Science* *((“long-term care” or “integrated care” or “home care”) and “systematic review”)* Stipulated time: Every year. Indexes: *SCI-EXPANDED, SSCI, A&HCI, CPCI-S, CPCI-SSH, ESCI, CCR-EXPANDED, IC.*
*Scopus* *(((“long-term care” or “integrated care” or “home care”) and “systematic review”)) AND (LIMIT-TO (DOCTYPE,”re”) OR LIMIT-TO (DOCTYPE,”ar”))*

**Table 2 ijerph-20-01698-t002:** Number of articles included in the mapping of the central concepts of the umbrella review.

Without Institutionalisation	
Family caregivers	
Quality Outcomes	7 articles included
Interventions	6 articles included
Family caregivers and patient	
Case Management	3 articles included
Quality Outcomes	3 articles included
Interventions	1 article included
Patients	
Case Management	4 articles included
Home hospitalisation	1 article included
Quality Outcomes	12 articles included
Interventions	19 articles included
System reform: cases	4 articles included
System reform: financial impact	3 articles included
Technology	7 articles included
Patients not walking without support/not getting out of bed	
From hospital to community	1 article included
Quality Outcomes	1 article included
Interventions	1 article included
Patients walking/leaving home	
From hospital to community	1 article included
Interventions	4 articles included
With Institutionalisation	
Family caregivers	
Quality Outcomes	1 article included
Family caregivers and patient	
Quality Outcomes	2 articles included
Interventions	2 articles included
Technology	1 article included
Patients	
From hospital to community	1 article included
Quality Outcomes	27 articles included
Interventions	80 articles included
System reform: cases	3 articles included
System reform: financial impact	6 articles included
Technology	3 articles included
Patients not walking without support/not getting out of bed	
Interventions	2 articles included
Mixed	
Family caregivers	
Quality Outcomes	5 articles included
Interventions	3 articles included
Family caregivers and patient	
From hospital to community	1 article included
Quality Outcomes	2 articles included
Interventions	2 articles included
System reform: financial impact	1 article included
Technology	1 article included
Patients	
From hospital to community	2 articles included
Case Management	2 articles included
Home hospitalisation	1 article included
Quality Outcomes	28 articles included
Interventions	52 articles included
System reform: cases	14 articles included
System reform: financial impact	18 articles included
Technology	10 articles included
Patients not walking without support/not getting out of bed	
System reform: financial impact	1 article included
Patients walking/leaving home	
Interventions	1 article included

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
