# Peer review of "Challenges to Systems of Long-Term Care: Mapping of the Central Concepts from an Umbrella Review"

_ijerph, 2023, doi:10.3390/ijerph20031698_

Round 1
Reviewer 1 Report
This study represents a massive effort and it deserves to be published somewhere, as these data will most likely continue to be a unique resource well into the future. This manuscript presents very interesting data on worldwide health systems for high aged population. The study provides an important contribution to building up the world-wide appropriate models for the needs of long-term care for older people.
This paper is overall descriptive and lacks in-depth discussions.
As the authors cited, regarding nursing care for the elderly, it may be necessary to consider the actual situation, including historical and religious background, education, economy, and language in each country. I think it would be necessary to fully understand the situation of each country before conducting an advanced analysis.
I recommend to revise the manuscript so that the purpose and result appear more clearly. Are the conclusions consistent with the evidence and arguments presented and do they address the main question posed? Simply, what is the most important thing or aspect for elderly care?
Author Response
Dear reviewer
First of all, we would like to thank you for your fantastic contributions in reviewing the article, which allowed us to improve the article a lot.
This revision of the manuscript allowed a deeper understanding of the results and the discussion, taking into account the purpose of the study and improving the conclusion. The manuscript also obtained significant clarifications in terms of method and inclusion and exclusion criteria.
Throughout the manuscript, the term "elderly" was replaced by "elderly".
The literature review included a clear definition of what long-term care for the elderly is and what it includes (lines 71 to 92);
In the Methods section, the inclusion and exclusion criteria have been expanded (lines 131 to 153);
A new Prisma Flow Diagram (page 7) was included, where the reasons for exclusion were included in the two stages, and table 3 was excluded;
Figure 2 was corrected because the 28 articles focusing on health professionals were excluded and referenced in PRISMA.
The results were more detailed and included the necessary explanation (lines 216 to 229).
Explained in the results because there is a higher incidence in nursing interventions (lines 334 to 346).
The discussion and conclusion section has been changed to better understand and systematize the knowledge generated by this review (Lines 503 to 548).
In the manuscript we attach, the changes are shaded in the document.
Yours sincerely
Luís Filipe Barreira

Reviewer 2 Report
The manuscript presents an umbrella study examining the various aspects of long-term care for older adults. This ambitious and broad study can significantly contribute to mapping existing research knowledge on this critical topic. However, there is a need for a more precise definition of long-term care, clarification of the research methodology and findings, and a more accurate explanation of the possible contribution of the research, as detailed below.
1. Throughout the manuscript, the term "elderly" is used. Since in the last decade, there has been criticism of the use of this term perceived as ageist, it is advisable to consider changing the term to "older adult" (see
Avers, D., Brown, M.map its diverse aspects. (2011). Use of the term “elderly”. Journal of Geriatric Physical Therapy, 34(4), 153-154.
2. There is a lack of a clear definition in the literature review of what long-term care for older adults is and what it includes. The review focuses mainly on the health aspect of long-term care (for example, lines 42-45). It is recommended to bring a clear definition to long-term care and to map the diverse aspects of this care.
3. In the Methods section, it is recommended to clarify and expand the inclusion criteria (lines 108-113). What is included in the definition of long-term care? Which kinds of medical interventions are included in this review? Why? In addition, what are the exclusion criteria?
4. In the Prisma Flow Diagram (page 7). It is recommended to clarify the reasons for excluding the articles in the two steps, as well as the difference between these two exclusions (besides the explanation in line 161 that the first exclusion included abstracts and titles, and the second exclusion included complete articles). In addition, the relationships between the two steps of exclusion in the diagram and the reasons for exclusion, detailed in table 3 (pages 7-8), need to be clarified.
5. In Figure 2- The conceptual mapping scheme (page 8). It needs to be clarified why the analysis of the findings regarding the context (with or without institutionalization or mixed) relates to 378 articles while the other links to 350 articles.
6. The results detailed extensively the eight categories that emerged from the patient perspective (lines 189-191). These categories are parallel to one of the sections in table 2 (page 6). However, a reference and explanation of the other sections of this table in the Result section need to be included.
7. Results chapter, line 192 - The definition of long-term care in the community is essential, but it is recommended to include it in the literature review. This is in addition to the recommendation to bring in the literature review a clear definition and description of the other aspects of long-term care and the general definition of long-term care.
8. Results chapter, line 291- It is recommended to explain why the reference is only for nursing interventions and not other kinds of interventions.
9. Lines 349-394- all these paragraphs repeat exactly the paragraphs in lines 301-345.
10. Following the recommendation to define and describe clearly in the literature review section the definition of long-term care for older adults and its different aspects and explain more systematically the research methods and the findings, it is recommended to write the discussion section in a more precise and focused manner. Now the results are comprehensive and varied, and it isn't easy to understand from this what is the message and the significant contribution of the article to the overall body of research knowledge of long-term care for older adults.
Author Response
dear reviewer
First of all, we would like to thank you for your fantastic contributions in reviewing the article, which allowed us to improve the article a lot.
This revision of the manuscript allowed a more precise definition of long-term care, deepening the results and discussion, taking into account the objective of the study. The manuscript also obtained significant clarification in terms of method and inclusion and exclusion criteria. All recommendations were important and incorporated in the reformulation of the manuscript.
Next, we identify the changes to the manuscript point by point:
1 - Throughout the manuscript, the term "elderly" was replaced by "older adult";
2 and 7 - A clear definition was included in the literature review on what long-term care for the elderly is and what it includes (lines 71 - 92);
3 - In the Methods section, the inclusion and exclusion criteria were expanded (lines 131-153);
4 - A new Prisma Flow Diagram (page 7) was included, where the reasons for exclusion in the two stages were included, and table 3 was excluded;
5 - Figure 2 was corrected because the 28 articles focusing on health professionals were excluded and referred to in PRISMA.
6 - The results were more detailed and included the necessary explanation (lines 216 - 229);
8 - Explained in the results why there is a higher incidence in nursing interventions (lines 334 - 346);
9 - Removed the repeated paragraphs that were included by mistake;
10 - The discussion and conclusion section were changed in order to better understand and systematize the knowledge generated by this review (Lines 503 - 548).
In the manuscript we attach, the changes are shaded in the document.
Thanks
Best regards
Luis Filipe Barreira

Round 2
Reviewer 2 Report
The authors have addressed all my previous issues. I have no further comments